# Does the Experience of Caring for a Severely Disabled Relative Impact Advance Care Planning? A Qualitative Study of Caregivers of Disabled Patients

**DOI:** 10.3390/ijerph17051594

**Published:** 2020-03-02

**Authors:** Fu-Ming Chiang, Jyh-Gang Hsieh, Sheng-Yu Fan, Ying-Wei Wang, Shu-Chen Wang

**Affiliations:** 1Department of Nursing, Institute of Medical Sciences, Tzu Chi University, Hualien Tzu Chi Hospital, Buddhist Tzu Chi Medical Foundation, Hualien 97002, Taiwan; mingfly6786@yahoo.com.tw; 2Department of Family Medicine, Institute of Health and Welfare Policy, National Yang-Ming University, Hualien Tzu Chi Hospital Buddhist Tzu Chi Medical Foundation, Hualien 97002, Taiwan; jyhgang@gmail.com; 3Institute of Gerontology, College of Medicine, National Cheng Kung University, Tainan 70101, Taiwan; shengyu@mail.ncku.edu.tw; 4Institute of Medical Sciences, School of Medicine, Tzu Chi University, Hualien 97004, Taiwan; 5Department of Family Medicine, Hualien Tzu Chi Hospital, Buddhist Tzu Chi Medical Foundation, Hualien 97002, Taiwan; 6Department of Nursing, Hualien Tzu Chi Hospital Buddhist Tzu Chi Medical Foundation, Hualien 97002, Taiwan; wangsc@tzuchi.com.tw

**Keywords:** advance care planning, caregiver, end-of-life decision

## Abstract

The aging of the Taiwanese population has become a major issue. Previous research has focused on the burden and stress faced by caregivers, but has not explored how the experience of these caregivers influences decisions of advance care planning (ACP). Semi-structured and in-depth interviews were conducted. Qualitative content analysis was used to identify important themes. Five themes and fourteen sub-themes were identified: (1) Past experiences: patient wishes, professional recommendations, and expectation about disease progress; (2) Impact of care on family members: positive affirmation, open-minded life, social isolation and health effects, and financial and life planning effects; (3) Attitude toward life: not forcing to stay, and not becoming a burden, (4) Expected proxy dilemmas: torment between doing or not, seeing the extension of suffering and toil, and remorse and self-blame; (5) Expectation of end of life (EOL) care: caregiver’s experience and EOL care decisions, and practicality of EOL decision making. After making multiple medical decisions for their disabled relatives, caregivers are able to calmly face their own medical decisions, and “not becoming a burden” is their primary consideration. It’s suggested that implementation of shared decision-making on medical care for patients with chronic disability will not only improve the quality of their medical care but also reduce the development of remorse and guilty feelings of caregivers after making medical decisions.

## 1. Introduction

With the increase of the elderly population, Taiwan became an aging society in 2018, as defined by the World Health Organization. Based on an estimation of the disability rate of 15.91%, the number of disabled elderly people that require assistance and care is around 0.55 million [1], of whom 65% are cared for by their family members [2]. According to survey results, every family caregiver spends around 10 h every day caring for disabled elderly family members and this lasted for around 9.9 years, resulting in an immense care burden [3]. During the entire course of the disease, family members experience the same crisis feelings as the patients, which affect their own attitude toward life and future [4]. Owing to various changes caused by suffering from relatives’ disease as they near death, caregivers tend to experience negative emotions such as anxiety, depression, and loneliness, and suffer from immense stress [5]. Studies found that two-thirds of caregivers complained of health problems such as sleep disorders, negative emotions like anxiety, depression and loneliness, and have poorer quality of life [6,7]. The study pointed out that care experiences and the experience of the death of relatives causes caregivers to think about the value and meaning of life [8], urging them to prepare for aging [9] and reexamine their value of life [10,11], which also brings spiritual burden [12].

Advance care planning (ACP) refers to a process of communication and discussion on the patient’s expected end-of-life care among the patient, family members, proxy decision-maker, and medical staff [13,14]. In addition to considering the patient’s autonomy and values [15], ACP undergoes a continuous communication, preparation, and reflective discussion process, which reduces the emotional stress in caregivers when caring the terminal patients [16]. Studies on caregivers found that most studies focused on caregiver burden, stress, needs, care knowledge and skills, or quality of life [17,18,19]. Most studies that examined end-of-life care decisions or medical willingness are mostly on whether the patient’s medical willingness is violated or adhered to, as well as their decision-making process and healthcare satisfaction. Studies examining the medical wishes or ACP of caregivers themselves after experiencing the care process are extremely rare. Dying is an inevitable process of life but there is a taboo on discussing death in traditional Chinese society. Due to caring for disabled relatives, family caregivers are accustomed to facing issues on the meaning of life earlier than the public. Therefore, there is a need for in-depth understanding about the care experience of caregivers and how it affects the attitude about their own ACP. The aim of this study was to explore the perception of ACP for caregivers of disabled critically ill relative and the effects of their care experience.

## 2. Materials and Methods

### 2.1. Participants and Design

Primary caregivers of long-term home care patients in eastern Taiwan were enrolled as subjects in this study. The caregivers recruited in this study must fulfill the following criteria: (1) Family members who previously received home care services. (2) The caregivers can communicate in Chinese or Taiwanese, and can clearly express themselves and communicate with the researcher. (3) Those who agreed to participate in this study after listening to the explanation of the researchers and simultaneously consenting to interviews and voice recordings. The aim of this study was to explore the perception of ACP for caregivers of disabled critically ill relative and the effects of their care experience. This qualitative study employs purposive sampling and in-depth interviews for data collection. The interview venue is the home of the caregiver. The researcher explained the aim and process of this study to the caregiver. The participation of this study was voluntary and written informed consent was signed by caregivers before data collection. On obtaining consent, the researcher conducted data collection according to a semi-structured interview guide. Recruitment of participants continued until data saturation, i.e., researcher noticed the occurrence of repetitive themes with no emergence of new data [20]. All of the interviews were conducted by the same researcher.

### 2.2. Data Collection and Analysis

One-on-one audio-recorded in-depth interviews were conducted from June to December 2014. The semi-structured interview included information about: (1) the situation of the disabled family member insertion of invasive tubing (nasogastric tube, urinary catheter, tracheostomy tube); (2) the personal feelings of the caregiver; (3) what are their thoughts when facing the same situation in the future? What do they expect of the surrogate decision-making? and (4) what will others value their decision? Before the end of the interview, the researcher made a conclusion on the interview content and requested the caregiver to review the researcher’s conclusion to see if it fully describes their feelings and whether additional information on ACP are needed. Within 24–48 h after the interview, the researcher transcribed the recording into a verbatim transcript and coded the data by herself. Following that, the researcher continuously read the content, reflected on it, and employed the “category-content” analytical model [21] for repeated analysis of the text. Major themes were summarized and categorized.

### 2.3. Methodological Rigor

Lincoln and Guba presented four criteria for examining the trustworthiness of qualitative studies [22]. To establish credibility, the researcher formed trusting relationships with the participants prior to the interviews and use peer debriefing and member checking to ensure authenticity. To establish transferability, open-ended questions were used to allow the participants for a comprehensive reflection of their experiences. When vague responses were given, the researcher encouraged and guided the participants to further describe the experience and clarify the meaning. To establish dependability, a qualitative expert was consulted throughout the process of data analysis. The original data and meaningful descriptions were presented to and discussed with peers to reach a consensus. To establish conformability, the researchers examined their own perceptions toward the research questions and continued to reflect on them throughout the research process to minimize any possible personal bias. The data obtained were marked with times and dates and properly stored for future reference. The researchers categorized similar contents into themes, analyzed the contents systematically, and provided thorough descriptions of the participants’ perspectives. The Standards for Reporting Qualitative Research (SRQR) checklist was followed to report the study [23].

### 2.4. Ethical Considerations

This study was reviewed and approved by the Institutional Review Board of the study hospital located in eastern Taiwan. The researcher explained the study aim and methods in detail to family caregivers who fulfilled the inclusion criteria. If the subject was willing to participate and complete the questionnaire, he/she was asked to complete the informed consent form before answering. They had the right to refuse to participate or to withdraw from this study.

## 3. Results

### 3.1. Sample Characteristics

We interviewed 12 caregivers (A–L) of disabled and critically ill patients and the interview data was saturated. The ages of the subjects aged 42–67 (mean age: 56.4) years; there were six males and six females. The mean caregiving duration was 6.2 years and most subjects have a senior high school education level (Table 1).

### 3.2. Themes

We used NVivo software (QSR International Pty Ltd., Daresbury, Cheshire, UK) to analyze the data to obtain five themes (1) Past experience; (2) Attitudes toward life; (3) Impact of caring for family members; (4) Expected proxy dilemma; and (5) Expecting end-of-life care (Table 2).

#### 3.2.1. Past Experience

The caregiver recollects suggestive language toward end-of-life care and acceptance toward invasive life support tube insertion. This theme includes three subthemes: patient’s wishes, professional recommendations, and expectation about disease progress.

##### Patient’s Wishes

Before losing consciousness, the patient revealed to the caregiver his thoughts on end-of-life care:

*“My dad previously stated that he wished to die in peace and not be a burden to his family.”* (Subject F)

##### Professional Recommendations

The medical professional recommended using invasive tubing in the patient:

*“We will do whatever the physician recommends. An example is tracheotomy, which is good for sputum suction according to the physician.”* (Subject L)

##### Expectation about Disease Progress

The physician suggested a poor prognosis of the patients, the caregiver still kept a glimmer of hope and requests the physician to try the best in treatment:

*“He is very young and our only son. Although the physician said that the prognosis is not optimistic, but we still requested that he tried his best with advanced medicine. There will be hope.”* (Subject E)

#### 3.2.2. Impact on Caregiver

From the time that the disabled family member is discharged, the caregiver must face the responsibility of caring for the patient alone and shoulder the burden of a “caregiving role.” During the predicament of gradually increasing caregiving pressure, caregivers experience impacts and changes in lifestyle, social activities, finances, and emotions and begin to have thoughts on the patient and current caregiving status. This includes four subthemes: positive affirmation, open-minded life, social isolation and health effects, and financial and life-planning effects.

##### Positive Affirmation

Caregivers affirmed their value through the praise and encouragement of people around them on their caregiving efforts:

*“She is difficult to care for and I often experience problems at the start. Subsequently, after solving (problems) for a long time, I know how to care for her better and everybody says that it is not easy for me.”* (Subject K)

##### Open-Minded Life

Caregivers became more tolerant and magnanimous, not nitpicking about immediate gain or loss, and not making things difficult for themselves and others after experiencing the caregiving process:

*“I felt that there is nothing to fight for and nothing is worth taking away. If there is really no hope, there is nothing I cannot let go. It is more painful if I cannot leave”* (Subject H)

##### Social Isolation and Health Effects

Caregiver experiences immense stress during caregiving and their pace of life and emotions change with the physical status of the patient. Although pain occurs in the patient, the caregiver also felt suffering and helpless:

*“At the start after my grandfather was discharged, me and my foreign maid do not know how to take care of him and will be flustered and nervous when there is any noise. We were unable to eat and sleep well. Everybody was suffering at home and our mental state was also poor! This was really terrifying as we had no one to turn to.”* (Subject A)

*“It is really tiring to pull this for a long period of time, although it has not caused pain. Of course, I hope to rest but there is no time for that. How am I supposed to take a rest?”* (Subject G)

##### Financial and Life-Planning Effects

Caregiver experiences limitations and burdens in finance and life development stages because of their caregiving duties:

*“To put it bluntly, he will not wake up and is just there consuming our pension, which is the reason why my wife cannot retire. Our burden is so huge and our granddaughter requires us to raise her. What shall we do?”* (Subject E)

#### 3.2.3. Attitudes toward Life

After the caregiver has experienced the actual caregiving process, they will reflect on the meaning and the value of life. While providing care, they will unintentionally think about the same thing happening to them in their future and their expectations and plans for care. This theme includes three subthemes: Not forcing to stay, not becoming a burden.

##### Not Forcing to Stay

When life reaches its end, it should be peaceful and not experience physical or psychological suffering. Caregivers mentioned that they do not wish to live with the invasive life support tubing in such situations:

*“I told my son to let me leave the world naturally if I am in a vegetative states so as not to see me suffer. My son should not intentionally keep me alive in such situation.”* (Subject A)

##### Not Becoming a Burden

Caregivers mentioned that they don’t want to become a burden on others and drag their family down in the event they lose their ability to care for themselves:

*“I felt that I should not be a burden to others. This is my personal thought, which differs from person to person. I felt that this burden should not be passed to others, including my son. Previously I told my son to let go if such a day occurs as I do not want him to suffer.”* (Subject D)

#### 3.2.4. Expected Proxy Dilemmas

As caregivers experienced the stress and burden of caregiving work, they worry that their children will experience the same fate of hard work and may face blame and difficulties. This theme encompassed three subthemes: torment between doing or not doing, seeing the extension of suffering and toil, and remorse and self-blame.

##### Torment between Doing or not Doing

When caregivers faced decisions making related to treatment choices for their family member, they were filled with uncertainty, hesitation, conflict, and other complex feelings:

*“It is a difficult choice to make! It is wrong to carry it out but it is also not right if it is not carried out. An example is when making a decision on tracheotomy on my grandfather when everybody has different opinions. Finally, I had to make the decision, which was really painful! I had led a hard life and it is tough living because I do not know what to do. In fact, there is no choice!”* (Subject A)

##### Seeing the Extension of Suffering and Toil

Most caregivers mentioned seeing the extension of suffering and toil as they “saw” the physical and mental anguish of disease experienced by their relatives. The helpless and cumbersome care causes them to feel that care is tough. The pain and torment suffered by both party causes them to hope that their children do not have to experience the role of a “caregiver”:

*“Life is tough by letting her live for one more year. Her quality of life is very poor and I am unable to shoulder her pain and breathlessness and cannot let her feel better. Caring is very tormenting and there is a lot of helplessness and pain. Although the child is pitiful, I am also pitiful.”* (Subject C)

##### Remorse and Self-Blame

Caregivers mentioned that as caregiving duration increases, they gradually understood that their patient’s condition was not what they expected. When they was no hope, they feel remorse and guilt about their initial decision:

*“It is useless being in a vegetative state, like my grandfather. I do not know how many families are burdened by such family members. What is the meaning of living like this? Wasteful, burdening family members, looking at him makes my day gloomy.”* (Subject A)

#### 3.2.5. The Expect of End-Of-Life (EOL) Care

Most caregivers expressed that the caring experience and process was tedious and tormenting. They would not want their loved ones or family to experience the same process when they get critically sick, they did not want to be a burden for their families. They had begun arranging their own end-of-life affairs to exterminate the fate of being cared for. This theme encompassed two subthemes: caregiver’s care experience and his/her own end-of-life (EOL) care decisions and practicality of one’s EOL decision-making.

##### Caregiver’s Care Experience and His/Her Own EOL Care Decisions

Most caregivers expressed that during the care process, they were always filled with uncertainty, hesitation, and ambivalence whenever having to make a choice for treatment related to the patient. The caring process is also very tiring indeed.

*“It is exhausting no matter what you do! It is wrong if you go ahead with it, but it also does not seem right to not do anything. Like the time when we had to decide whether grandpa should undergo tracheostomy. Everyone had different opinions, and at the end, I was still the one to make the decision. Like me, I do not even know whether it was the right thing to do or not…”* (Subject A)

##### Practicality of One’s EOL Decision-Making

Caregivers expresses either verbally or on the National Health Insurance (NHI) card that he/she does not want to be resuscitated so that the next generation would not go through the same fate he/she did.

*“I think that I do not want to become a burden on someone. That is my belief, and everyone has different beliefs. I believe I should not put such a burden on somebody else, even if it is my own child. I already had this conversation with my son a long time ago—if the day comes, just let it go. It has been noted in the NHI card already—do not resuscitate.”* (Subject K)

## 4. Discussion

Our study found in exploration of caregiver ACP that “not becoming a burden” is a common core theme that many caregivers of critically ill and disabled patients share. The caregivers mentioned that if they lose consciousness, they are akin to being dead and not becoming a burden to others. To avoid becoming a burden to their family members like their patient, the caregivers are more willing to discuss and even actively plan or sign advance directives for their EOL care wishes, and express their wishes to their immediate family members or medical staff. Scholars mentioned that it is an extremely stressful and burdensome process for family members to speculate whether the patient has a do not resuscitate order or life-sustaining therapy decisions [24]. Through the ACP process, family members will be better able to understand the patient’s wishes, values, and EOL care decisions, assisting them in having a peaceful death, and thus feeling consoled. At the same time, family members can prepare for the death and loss of patients, which decreases remorse and alleviates stress in family members due to making decisions when they do not know the patient’s wishes. This also avoids doubts and conflicts between family members with differing opinions and protects them from accusations and attacks from other family members after the patient has died [25]. The interviews found that caregiving experience drives caregivers to think deeply about life dignity and EOL issues. Five caregivers mentioned that they hope to “leave peacefully,” “leave without suffering, “let nature takes its course,” and “leave with dignity,” which is consistent with previous studies [26].

From an ethics perspective, the patient’s wishes are the most important premise when making decisions. If the patient’s wishes are unclear, the patient’s preferences and consideration of the patient’s maximum benefits are priorities for decision-making [27]. Due to the effects of traditional values on filial piety, the young generation are fear to be branded as unfilial, spouses are fear to be accused of being irresponsible, and most caregivers request the physician to try their best to save the patient. Factors such as not being able to bear with the thought of losing the patient causes the patient’s wishes, preferences, and interests to not be considered as priorities in making medical decisions. When faced with making decisions, caregivers often hesitate and struggle between “other family members’ wishes” and “patient’s wishes,” of which “patient’s wishes” is often the losing side. These results are also similar to those found by other researchers [28,29]. From this, we can see that nationality and culture are not absolute factors. The system of appointing a “medical proxy” is to ensure that the wishes of terminal patients and patient autonomy can be maximally protected. Researchers recommend that patients should have sufficient communication with family members when they are conscious, regardless of whether they have a designated proxy [30]. Taiwan’s medical environment has put more emphasis on the promotion of shared decision-making in recent years. It is to protect the patient’s right to autonomy and to decrease the caregiver’s pressure on caregivers to make medical decisions and chances for regret and self-blame afterwards.

A series of care problems and impact on family caused by bedridden and disabled family members is usually more than what the caregiver expects. Studies have shown that nearly one-third of the proxies developed negative emotions such as guild, stress, anxiety, and self-doubt after deciding the treatment for their family members, which persists for several months or even several years [31,32]. Three caregivers participated in the study mentioned that there is no dignity and meaning in living in pain like their patients. When family members receive invasive life support tubing, their life as well as their suffering is extended, and the source of this suffering is caused by the invasive life support tubing. Examples include pain from sputum suction, insertion of tubes, and being restrained. Seeing the discomfort in their family members due to suffering further strengthens their negative thoughts that “living is suffering” and tends to cause caregivers to feel guilty and depressed, which is similar to the results of past studies [33,34]. Conversely, if the patient shows peaceful countenance with absence of pain, the caregiver will instead feel positive. Some caregiver thought mentioned that the nasogastric tube was inserted to prevent her mother from choking, and this is a form of repayment to her mother for bringing her up. Although the 10 years of caring is tough, she had positive attitudes toward it, which is consistent with previous studies [35].

Caregivers cannot bear to see their family member suddenly die and thus selected aggressive treatment to save the patient. However, this resulted in conflict on care issues and they felt angry and disappointed with the uncaring attitude of other family members. The literature has mentioned that caregivers tend to experience caregiver burden when others do not share their caregiving work and family members do not cooperate and support them [36]. Researchers mentioned that caregiver burden occurs when physiological, psychological, emotional, social, and financial problems occur while caring for disabled family members [37]. When caregiver burden is greater than care in caregivers, the emotional connection will gradually disappear and caregiving will bring out many changes and stress in life [38]. We acknowledge limitations to our study. Firstly, the study was conducted at a single home care institution. Secondly, geographical location of participants may influence the generalizability of the results. However, the study provided valuable information of the meaning of life for caregiver after taking care of severely disable patients.

## 5. Conclusions

This study is the first to explore the effects of caregiving experience on ACP in caregivers. During interviews, we found that most caregivers experienced and thought about the value and meaning of life during caring. After experiencing making multiple medical decisions for their disabled relatives, caregivers are able to calmly face their medical decisions and “not becoming a burden” is their primary consideration. Studies have found that family caregivers, compared with the general public, are more positive and willing to accept the concept of ACP. Caregivers’ past experiences, attitudes toward life, impact on the caregiver, expected proxy dilemmas are important factors for promoting caregivers’ views of EOL care. The German philosopher, Martin Heidegger, once mentioned, “Birth is the start of life and death is the common endpoint for everyone, the process of life is also the process of death.” Life is even tougher for caregivers due to their caregiving experience. It is expected that sound communication, provision of appropriate and correct AD instructions, and respect for EOL autonomy can make terminal life more fulfilling and end in peace without regrets.

## Figures and Tables

**Table 1 ijerph-17-01594-t001:** Characteristics of the A~L caregivers (*N* = 12).

Item		*N*	%
Gender	Male	6	50
	Female	6	50
Caregiver Age	≤49	2	16.8
	50–59	5	41.6
	60–69	5	41.6
Patient Age	≤49	4	33.4
	50–69	1	8.3
	70–89	6	50
	≥90	1	8.3
Caring period(year)	≤1	1	8.3
	>1–6	6	50
	>6–9	3	25
	≥10	2	16.7

**Table 2 ijerph-17-01594-t002:** Summary of themes and sub-themes.

Theme	Sub-Theme
(1) Past experience	Patient wishes, Professional recommendations, Expectation about disease progress
(2) Impact on caregiver	Positive affirmation, Open-minded life, Social isolation and health effects, Financial and life-planning effects
(3) Attitudes toward life	Not forcing to stay, Not becoming a burden
(4) Expected proxy dilemma	Torment between doing or not doing, seeing the extension of suffering and toil, Remorse and self-blame
(5) The expect of end-of-life (EOL) care	Caregiver’s care experience and his/her own EOL care decisions, Practicality of one’s EOL decision-making

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
