# Peer review of "Does the Experience of Caring for a Severely Disabled Relative Impact Advance Care Planning? A Qualitative Study of Caregivers of Disabled Patients"

_ijerph, 2020, doi:10.3390/ijerph17051594_

Round 1

Reviewer 1 Report

Manuscript Number: ijerph-713162

Full Title: Does the experience of caring for a severely disabled relative affect the decision of advance care planning?

Thanks for having asked me to review this qualitative study focused on exploring how informal caregivers’ experiences of disabled critically ill patients affects caregivers’ attitude about their own advance care planning.

I have some comments that should be addressed to improve the manuscript.

Title

The title does not clearly reflect that you are making a qualitative study and participants are caregivers of disabled patients.

Materials and Methods

Page 2, line 76, I wonder if along the informed consent process the researcher explained the study procedures in addition to the study aim.

Page 2, line 77 I suggest modifying the line about informed consent; from “sought consent” to “participation was voluntary and written informed consent was signed by caregivers before data collection.”

Page 2 and 3 The authors did not report study procedures in detail. It would be worthy to the readers include the procedure used to identify caregivers, and specify more how the researcher selected the participants for recruitment?

It is not clear when the study was conducted. Please clarify this.

Results

It is not clear which reporting guideline the authors used for reporting of qualitative research. I suggest when revising your manuscript, that your chosen EQUATOR checklist has been declared and cited properly.

Table 1. It seems that reports characteristics both of caregivers and patients, please modify accordingly.

Table 1 Column No: What does “No” stand for?

Table 1 Column “past history” Is the past history of caregivers or past history of patients?

Table 1 “Care years” is the number of years of caregiving delivered to patients?

Table 1 “Patient age” is the patient age reported in years?

Table 1 “past history” is the patient past history? Or is the caregiver past history?

Author Response

Dear Reviewer,

Good Day!According to your comments, we have revised our manuscript ‘Does the experience of caring for a severely disabled relative impact advance care planning? A qualitative study of caregivers of disabled patients’, and now submitting again to International Journal of Environment Research and Public Health to be considered for publication.  The detailed responses are as follows.

Reviewer 2 Report

This paper presents important information on the experience of caregiving for a severely disabled relative and its impact on advance care planning. It presents data on a very important issue. Additional information on the methods used for data analysis needs to be included as well as overall revisions for spelling and grammar. The use of an editor may be most helpful. It appears that different styles of writing were used in different parts of the paper. This made the paper challenging to follow in some sections than others. There is one additional major concern that warrants being addressed throughout the paper. This reviewer is concerned that the way these data from such a small sample of participants is currently presented, it does not adequately protect the patient and caregiver participants. Please see additional specific comments/suggestions below:

Title suggestion: Because advance care planning already implied decision making, the title can be revised as follows, “Does the experience of caring for a severely disabled relative impact advance care planning?”

Materials and Methods:

  • Line 70: The term, “clear consciousness” should be restated for clarity. Does this refer to having the capacity to make decisions or being without cognitive impairment?
  • How many members of the research analyzed these data?
  • How were conflicts in coding resolved?

Ethical Considerations:

  • To protect the research participants. It is recommended that the IRB number is removed and the region of the location of the hospital is mentioned instead of the specific name of the hospital - to protect the identity of this small sample.

Table 1:

  • Because this sample size is so small, and all data were collected from one site (hospital), this reviewer is concerned that the way the demographic information is presented may not sufficiently protect the identities of the participants. It is strongly advised that this table is presented in the format of standard demographic tables using means and standard deviations, etc.

Table 2:

  • Please revise for spelling i.e. the word “Them” instead of “Theme”
  • The decision to use uppercase and lowercase is unclear. Please revise for consistency.

Themes:

  • The way that the themes and subthemes are currently arranged makes it a challenge to follow. Since the table clearly delineates themes and sub-themes, it is suggested that each theme is presented as a subheading and each of the subthemes can be contained in a paragraph or two under each theme. As currently written where a statement is provided followed by a quote is insufficient to provide an in-depth understanding of the result.
  • The references that follow the quotes, should be streamlined, again, to protect the identities of the participants. It could simply state P1 or something similar to represent Participant 1, etc.
  • Lines 129-131 are unclear. Please clarify.
  • The same is try for lines 137-140. Please revise for clarity.
  • The repeated reference to “It’s refer” or “It’s mean” is not grammatically correct. Please revise
  • Line 227: It is not clear why “end-of-life” followed by “EOL” is spelled out here when the acronym was already used in the paper prior to this point.

Discussion:

  • Great discussion points made. Again, it is recommended that the detailed references to each study participant are removed. For example, on page 284, it is just as powerful to state that “Three caregiver participants in this study……” instead of identifying them as participants A, G, and E.

Author Response

(The authors gave the same response as above.)
